# A Synthetic Pathway for the Production of Benzylsuccinate in *Escherichia coli*

**DOI:** 10.3390/molecules29020415

**Published:** 2024-01-15

**Authors:** Johanna Mock, Karola Schühle, Uwe Linne, Marco Mock, Johann Heider

**Affiliations:** 1Fachbereich Biologe, Philipps-University Marburg, Karl-von-Frisch-Str. 8, 35043 Marburg, Germany; 2Synmikro Center Marburg, Karl-von-Frisch-Str. 8, 35043 Marburg, Germany; 3Fachbereich Chemie, Philipps-University Marburg, Hans-Meerwein-Str. 10, 35043 Marburg, Germany

**Keywords:** anaerobic toluene degradation, reverse β-oxidation, synthetic pathway, transport, CoA ligase, CoA-transferase, benzylsuccinate

## Abstract

(*R*)-Benzylsuccinate is generated in anaerobic toluene degradation by the radical addition of toluene to fumarate and further degraded to benzoyl-CoA by a β-oxidation pathway. Using metabolic modules for benzoate transport and activation to benzoyl-CoA and the enzymes of benzylsuccinate β-oxidation, we established an artificial pathway for benzylsuccinate production in *Escherichia coli*, which is based on its degradation pathway running in reverse. Benzoate is supplied to the medium but needs to be converted to benzoyl-CoA by an uptake transporter and a benzoate-CoA ligase or CoA-transferase. In contrast, the second substrate succinate is endogenously produced from glucose under anaerobic conditions, and the constructed pathway includes a succinyl-CoA:benzylsuccinate CoA-transferase that activates it to the CoA-thioester. We present first evidence for the feasibility of this pathway and explore product yields under different growth conditions. Compared to aerobic cultures, the product yield increased more than 1000-fold in anaerobic glucose-fermenting cultures and showed further improvement under fumarate-respiring conditions. An important bottleneck to overcome appears to be product excretion, based on much higher recorded intracellular concentrations of benzylsuccinate, compared to those excreted. While no export system is known for benzylsuccinate, we observed an increased product yield after adding an unspecific mechanosensitive channel to the constructed pathway.

## 1. Introduction

(*R*)-Benzylsuccinic acid, a natural product originally identified as a carboxypeptidase inhibitor [1,2], was later identified as the first intermediate in the anaerobic degradation of toluene [3,4,5,6]. Accordingly, benzylsuccinate and related compounds generated from other hydrocarbons have been detected in contaminated anaerobic environments such as groundwater aquifers or sediments and are used to evaluate the extent of both contamination and bioremediation [7,8,9]. Chemical synthesis of benzylsuccinate has been reported by relatively complex procedures of four or more steps, which yield the compound either as a racemate or as highly enriched (*R*)- or (*S*)-enantiomers [10,11,12], which are commercially available for prices starting about 1000-fold higher than those of the starting substrates benzoate or succinate. Biological activity depends on the particular enantiomers, as only (*R*)-benzylsuccinate is produced during anaerobic toluene degradation or is active as a carboxypeptidase inhibitor, whereas (*S*)-benzylsuccinate is a specific building block for the diabetes drug mitiglinide [13]. Benzylsuccinate has also been reported as a component of a butylene succinate co-polymer with interesting characteristics [14]. Thus, there are prospects for a potential industrial application of the benzylsuccinate enantiomers, which may be enhanced by establishing a biotechnological process capable of a more convenient and enantiospecific synthesis.

In nature, (*R*)-benzylsuccinate is formed during anaerobic toluene degradation as an obligate intermediate by all known bacteria, which couple this metabolic pathway to anaerobic respiration with nitrate, sulfate, metal ions or protons, or to anoxygenic phototrophy [4]. Remarkably, the same principal reactions and highly similar orthologues of all the enzymes are used throughout by bacteria of very different taxonomic groups [4]. The pathway is initiated by the highly unusual stereospecific addition of the methyl group of toluene to the double bond of a fumarate cosubstrate to generate (*R*)-benzylsuccinate [15,16]. This reaction is catalyzed by the glycyl radical enzyme benzylsuccinate synthase, a highly oxygen-sensitive enzyme that needs to be activated to the active glycyl radical state by a separate S-adenosylmethionine-dependent activating enzyme [17,18]. The reaction has been shown to involve the addition of a benzyl radical generated from toluene to the *Re-Re* face of a bound fumarate and the return of the initially abstracted hydrogen atom to the resulting product radical in a *syn* addition mode, as well as a steric inversion of the configuration of the methyl group of toluene [4,6]. All these mechanistic details are fully consistent with the predicted reaction mechanism from quantum mechanics modeling [19] based on the X-ray structure of the enzyme [20,21].

Further degradation of (*R*)-benzylsuccinate proceeds via β-oxidation to benzoyl-CoA and succinyl-CoA in five subsequent enzymatic steps [22]. This pathway is initiated by benzylsuccinate CoA-transferase (BbsEF), which uses succinyl-CoA as CoA-donor for activating the intermediate to 2-(*R*)-benzylsuccinyl-CoA [15,23], and this is then dehydrogenated to (*E*)-benzylidenesuccinyl-CoA by benzylsuccinyl-CoA dehydrogenase (BbsG) [24,25]. The next steps consist of hydration of the double bond by benzylidenesuccinyl-CoA hydratase (BbsH) and another dehydrogenation of the alcohol intermediate by (*S*,*R*)-2-(α-hydroxybenzyl)succinyl-CoA dehydrogenase (BbsCD) [26]. Finally, the resulting 3-oxoacyl-CoA intermediate (*S*)-benzoylsuccinyl-CoA is thiolytically cleaved to benzoyl-CoA and succinyl-CoA by the thiolase BbsAB [27] (Figure 1). Succinyl-CoA is used as CoA-donor for activating benzylsuccinate, and benzoyl-CoA is further degraded by reducing the aromatic ring via benzoyl-CoA reductases [28]. All enzymes involved in anaerobic toluene degradation have been biochemically characterized and while the benzylsuccinate synthase reaction appears to be thermodynamically irreversible [4,6,19,29], all reactions of the enzymes of the β-oxidation pathway have been shown to be reversible [23,24,26,27]. Moreover, the enzymes and their gene organization in two co-induced operons are completely conserved in all known anaerobic toluene-degrading bacteria [4]. These operons are the *bss* operon containing the genes for the subunits of benzylsuccinate synthase (BssABC) and its activating enzyme (BssD), and the *bbs* operon containing the genes for the enzymes of benzylsuccinate β-oxidation.

In this communication, we establish a synthetic pathway for benzylsuccinate production in *Escherichia coli*. Because of the extreme sensitivity of benzylsuccinate synthase against oxygen, we attempted to produce benzylsuccinate via the degradative β-oxidation pathway running in reverse. To achieve this, we established one plasmid coding for a metabolic module for the uptake of benzoate and its activation to benzoyl-CoA and combined it with a second co-expressed plasmid containing the *bbs* genes. We indeed observed the production of benzylsuccinate under different growth conditions, especially in anaerobic glucose-fermenting cultures, which provide the cosubstrate succinate as a fermentation product.

## 2. Results

### 2.1. Biosynthetic Modules for Benzoyl-CoA Synthesis

#### 2.1.1. Benzoate Transport

To achieve the synthesis of benzoyl-CoA as a metabolic intermediate in *E. coli* cells, they first needed to be equipped with appropriate uptake and activation systems. Therefore, we cloned the *benK* gene (*ebA5311*) for a putative benzoate uptake carrier from the benzoate-degrading species *Aromatoleum aromaticum* [30] into the anhydrotetracycline-inducible broad host range vector pASG_mod via Stargate-cloning [31,32]. The resulting plasmid was transformed into strain Rosetta (DE3) pLysS, and the cells were grown on an LB medium and induced by adding AHT [32]. Although no induced protein was detected by analyzing the protein patterns of whole-cell-lysates by SDS-PAGE, we tested for the presence of a potential benzoate uptake system by analyzing for growth inhibition of the cells by adding benzoate to the medium. While the control strain without the plasmid showed consistent growth rates of 0.69 to 0.81 h^−1^ in the absence and up to 10 mM of added benzoate, the growth rates of the *benK*-containing strain decreased from 0.61 h^−1^ in the absence of benzoate to 0.21 h^−1^ in the presence of 10 mM benzoate (Figure 2). This clearly indicates that the transporter is active and that its presence leads to an accumulation of inhibitory benzoate concentrations in the cytoplasm. Growth inhibition by high benzoate concentrations was already observed before *benK* expression was induced and did not significantly change after adding AHT, suggesting that there is significant background expression of *benK* from the plasmid used.

#### 2.1.2. Benzoate Activation by CoA Ligase

After the uptake of benzoate into the cytoplasm, it needs to be activated to benzoyl-CoA, using either a benzoate-CoA ligase [33,34] or a CoA transferase [35]. Because there are known genes for either functionality in the genomes of toluene-degrading bacteria, we cloned the *bclA* gene (*ebA2757*) of *A. aromaticum*, coding for a benzoate-CoA ligase [30] and the *bct* gene (*Gmet_2054*) of *Geobacter metallireducens*, coding for a succinyl-CoA:benzoate CoA-transferase [35], into expression vectors. The *bclA* gene was cloned into vector pPSG5 (IBA, Göttingen) behind a T7 promotor and with an N-terminal streptag II and expressed in the *E. coli* strain Rosetta (DE3) pLys by induction with AHT. The protein was produced in large amounts, as illustrated by the presence of enzyme activity and a strongly induced protein band at 57 kDa after analyzing the cells by SDS-PAGE (Figure A1). Because purification attempts via streptactin affinity chromatography did not succeed, we purified the protein by sequential ammonium sulfate precipitation, desalting via a PD10 column, and anion exchange chromatography on an UnoQ column. The purification resulted in a 16-fold enrichment of the activity, yielding 2.6 mg of virtually pure protein with a specific activity of 107 U mg^−1^ (Table 1), which is significantly higher than those reported for other orthologous enzymes [33,34]. The enzyme was rather specific for benzoate and only accepted 2-fluorobenzoate as an alternative substrate with high activity (94% vs. benzoate), whereas it showed only marginal activity with 2-amino- or 4-hydroxybenzoate (1–2% vs. benzoate) and no detectable activity with 2-hydroxybenzoate.

#### 2.1.3. Benzoate Activation by CoA-Transferase

An alternative pathway for activating benzoate is represented by a succinyl-CoA:benzoate CoA-transferase present in the Fe(III)-reducing toluene- and benzoate-degrading species *Geobacter metallireducens* [35]. Therefore, we cloned the respective *bct* gene from this organism into broad host range expression vector pASG_mod [32], which provided an N-terminal streptag fusion, and produced the protein in *E. coli* strain Rosetta (DE3) pLys. The enzyme was purified by affinity chromatography using a streptactin column, resulting in 1.2 mg purified enzyme from a culture volume of 2 l (Figure A2). Enzyme activity was qualitatively tested by an HPLC-based assay, which was started by adding the substrates, succinyl-CoA and benzoate, followed by 10 min incubation at 30 °C. We observed the production of benzoyl-CoA at the expense of succinyl-CoA, indicating that the enzyme is sufficiently active to use it in the intended pathway (Figure 3). Because both benzoate-activating enzymes were obtained in active form, we continued to set up two separate pathways, one using the CoA ligase and the other using the CoA-transferase for benzoate activation.

#### 2.1.4. Combinations of Benzoate Transport and Activation Enzymes

To allow co-expression of the *benK* gene with either *bclA* or *bct*, we combined the genes in a vector compatible with the one chosen for cloning the *bbs* genes (see below). The genes were combined using the vectors pNFUSE and pCFUSE of the Stargate combinatorial cloning system (IBA, Göttingen, Germany), and the resulting synthetic operons were transferred into expression vector pPSG5 which contains a chloramphenicol resistance gene instead of the usual ampicillin resistance gene of vector pASG5. In several initial attempts, we never obtained correct clones containing either the *bclA* or the *bct* gene together with the *benK* gene. Regarding the previously observed growth inhibition by BenK even without induction, we assumed that the combined benzoate transporter and activating enzyme activities may deplete the cytoplasmic CoA pool by converting small benzoate amounts contained in the rich media to benzoyl-CoA. To counteract this effect, we applied another round of combinatorial cloning to additionally add the *bbsABCD* genes of *G. metallireducens* (*Gmet_1528*–*1531*) to the plasmids. These genes code for benzoylsuccinyl-CoA thiolase and (α-hydroxybenzyl)succinyl-CoA dehydrogenase (Figure 1) and were intended to be added anyway to create the synthetic pathway. The enzymes derived from these genes should convert benzoyl-CoA with endogenous succinyl-CoA to benzoylsuccinyl-CoA and (α-hydroxybenzyl)succinyl-CoA. These thioesters are much more sensitive to non-enzymatic hydrolysis than benzoyl-CoA [26,27], which should enhance CoA recycling. Using this modification, we indeed succeeded in cloning synthetic operons containing the *bclA* or *bct* genes together with the *bbsABCD* and *benK* genes (Figure 4). We confirmed the activities of the recombinant benzoate-activating enzymes resp. transporters produced from the combined plasmids using the same assays as described above.

### 2.2. Benzylsuccinate Synthetic Module

To achieve the conversion of benzoyl-CoA to benzylsuccinate, we intended to use the reverse-running β-oxidation pathway of benzylsuccinate to benzoyl-CoA, which participates in anaerobic toluene degradation. Because we intended to couple the reaction with anaerobically grown *E. coli* cultures producing succinate as a fermentation product [36,37,38], we decided to use the *bbs* operon of the ferric-iron-reducing bacterium *G. metallireducens* (genes *Gmet_1521–1531*) [39] as the source for the enzymes. Compared to denitrifying toluene-degrading bacteria, the *bbs* operons of *G. metallireducens* and other strictly anaerobic toluene degraders contain additional genes for the subunits of a special electron-transfer flavoprotein (ETF; *Gmet_1525* and *1526*) which serves as an electron acceptor for benzylsuccinyl-CoA dehydrogenase [25] and an ETF:menaquinone oxidoreductase (*Gmet_1527*) [40], which may result in a better performance of the pathway under anaerobic conditions. We succeeded in cloning the entire *bbs* operon of *G. metallireducens* into the broad-host-range plasmid pBBR2 [41] resulting in plasmid pBeta, which is compatible with the expression plasmids used for the genes for benzoate uptake and activation in terms of replication origins and antibiotics resistance genes (Figure 4).

### 2.3. Benzylsuccinate Production Assays

#### 2.3.1. Aerobic Conditions

Double transformants of *E. coli* strain DH5α with either pLigBen and pBeta or with pTransBen and pBeta were inoculated in either autoinduction (rich) medium or M9 minimal medium and tested for the production of benzylsuccinate under aerobic growth conditions. Both media were supplemented with 10 mM succinate and 2 mM benzoate to provide the starting substrates for the synthetic pathway. After incubation at 15 °C for three days, the cultures were centrifuged and the supernatant was extracted with ethylacetate. The concentrated extracts were then probed for the presence of benzylsuccinate by HPLC-MS analysis. To allow quantitative analysis, a small amount of phenylsuccinate was added to every supernatant before extraction, which served as an internal standard to compensate for variations in benzylsuccinate recovery between different samples. Phenylsuccinate is not known to occur in nature, but the two compounds are chemically highly similar and can still readily be distinguished by the different masses (see Figure A3). Using a benzylsuccinate standard solution spiked with phenylsuccinate, we obtained a linear calibration curve with a detection limit of around 10 nM after calibrating the integrated peaks with the internal standard. Because the extraction procedure provided a 50-fold accumulation of benzylsuccinate, we were able to assess supernatant concentrations down to 0.2 nM. The experiments in the autoinduction medium showed indeed the production of small amounts of benzylsuccinate, which amounted to 1.6 nM for the cells containing the CoA ligase, and 0.2 nM for those containing the CoA-transferase (Figure 5A). Although these values are close to the detection limit, we can regard them as significant since no detectable product was observed in the negative controls, either from cells lacking the plasmids or from an experiment without added benzoate (Figure 5A). The experiments in the M9 medium showed an accumulation of 0.5 nM benzylsuccinate for the cells containing benzoate-CoA ligase and 3.5 nM for those containing the CoA-transferase (Figure 5A).

#### 2.3.2. Anaerobic Conditions

Because succinate is among the usual metabolic products of *E. coli* cells under glucose-fermenting conditions, we tested whether the constructed plasmids enable it to produce benzylsuccinate without supplying relatively expensive succinate to the medium. Therefore, we set up cultures of recombinant cells carrying either pLigBen and pBeta or pTransBen and pBeta either in M9 minimal medium or in buffered rich medium (TGYEP) containing 10 mM glucose and 2 mM benzoate, but no succinate. Remarkably, the strain containing benzoate-CoA ligase produced 0.4 µM benzylsuccinate in the supernatant of the minimal medium and 2.5 µM in the rich medium, whereas the one containing the CoA-transferase was at the detection limit in minimal medium and reached 0.1 µM in rich medium (Figure 5B). Thus, the shift to anaerobic growth resulted in an almost 1000-fold increase of the maximum recorded yield of benzylsuccinate for the strain containing pLigBen and pBeta, even without the need to feed succinate to the culture. Because this strain produced 20–30-fold higher product yields than the one containing pTransBen and pBeta, the latter was discarded for further characterization and optimization attempts.

#### 2.3.3. Fumarate Respiration

Since succinate is a more abundant product under fumarate-respiratory than under fermentative conditions in *E. coli*, we compared the efficiency of benzylsuccinate production in M9 minimal media supplied with 10 mM glucose and 2 mM benzoate under these conditions, adding 40 mM fumarate as external electron acceptor. Using a minimal medium for these tests was intended to provide more controlled conditions, especially to exclude the presence of fumarate or its metabolic precursors such as aspartate or malate provided by components of the rich medium. The results showed benzylsuccinate concentrations of 0.5 µM in the supernatant of the fermentative culture, which were almost identical to those in the previous experiments in a minimal medium, whereas the fumarate-respiring culture yielded 4.8 µM, which corresponds to an increase by one order of magnitude (Figure 6A).

#### 2.3.4. Effect of Product Excretion

The results so far have been only evaluated by measuring benzylsuccinate concentrations in the culture supernatants. However, since no excretion system is known for this metabolite, productivity is expected to be limited by the ability of the cells to transport it out of the cell. Therefore, we investigated the amount of benzylsuccinate still present in the cells after the anaerobic production assays in the presence and absence of fumarate were completed. After harvested cells were washed to reduce the contamination by the remaining growth medium, the internal phenylsuccinate standard was added and the resuspended cells were lysed by adding TFA and incubating for 10 min at 60 °C. After extracting the organic acids from these samples by ethyl acetate extraction, the residue was analyzed by HPLC-MS analogously to the supernatant samples, yielding intracellular benzylsuccinate concentrations of 20 µM under fermentative and 57 µM under fumarate-respiring conditions at the end of the experiments (Figure 6B). Thus, the intracellular concentrations were more than 10-fold higher than those observed in the supernatants, which indicates problems in the excretion of the metabolite.

In order to alleviate the excretion of benzylsuccinate, we added the gene for a mutant derivative of a mechanosensitive channel of *E. coli* (*mscS* L09S) as an additional gene into vector pLigBen (Figure 4). The mutation causes a “gain-of-function” phenotype and results in a higher probability of opening the channel [42]. The resulting plasmid pLigBenEx was transformed into *E. coli* together with pBeta, and the effects on benzylsuccinate production under fermentative and fumarate-respiratory conditions were examined. As shown in Figure 6A, benzylsuccinate in the culture supernatant increased 3.5-fold under fermentative conditions but decreased 1.5-fold under fumarate-respiring conditions. The intracellular benzylsuccinate concentrations were still about 10-fold higher than those excreted into the medium and showed significant reduction only in the fumarate-respiring culture. These observations may be explained by a stronger negative effect of the mutant channel on fumarate respiration, which relies much more on an intact cytoplasmic membrane than fermentation.

## 3. Discussion

Synthetic pathways for producing benzylsuccinate can either be envisaged starting from toluene and fumarate or from benzoate and succinate. Although the sole natural pathway involving this intermediate, anaerobic toluene degradation, starts with the former reaction, the extreme oxygen sensitivity of benzylsuccinate synthase severely restricts its biotechnological applicability [4]. In addition, toxicity effects by the required amounts of toluene needed for decent product yields and the relatively high costs for the fumarate cosubstrate pose additional problems for this approach. Therefore, we constructed a pathway for benzylsuccinate synthesis from succinate and benzoate, employing the enzymes of β-oxidation of benzylsuccinate in reverse [22]. Since we have previously purified and characterized all five of these enzymes and found them all principally reversible [23,24,26,27], we regarded this approach as feasible. The only β-oxidation enzyme causing potential thermodynamic problems is benzylsuccinyl-CoA dehydrogenase (BbsG) because it delivers the electrons to the quinone pool via an electron transfer flavoprotein (ETF) and an ETF:quinone oxidoreductase [24,25]. The in vitro assays were performed with artificial electron carriers, ferricenium as the electron acceptor for the forward and benzyl viologen as the electron donor for the reverse reaction [25]. These reaction conditions have been designed to pull the reaction in the desired direction, but do not represent the natural situation. Moreover, the electron transfer cascades from BbsG to the quinones appear to differ slightly depending on the physiology of the host bacteria: denitrifying species carry only single gene copies in their genomes encoding either the ETF protein or the ETF:quinone oxidoreductase available for interacting with BbsG and transferring the electrons into the ubiquinone pool [30], while in ferric iron or sulfate-reducing bacteria, multiple copies of ETF proteins and associated ETF:menaquinone oxidoreductases (EMO; *orfX* gene products, Figure 4) are encoded in the operons of β-oxidation enzymes [39,43]. The different types of quinones in denitrifying vs. strictly anaerobic bacteria appear to be important, since they have quite different midpoint potentials (−80 mV for menaquinone, +110 mV for ubiquinone [44]), with menaquinol as a thermodynamically more favorable electron donor for the reverse-running pathway. The intended host species *E. coli* is known to shift its quinone pool from ubiquinone under aerobic and nitrate-respiratory conditions to menaquinone during fumarate respiration and fermentation [45]. Because we intended to use the mixed-acid fermentation pathway of *E. coli* to produce the cosubstrate succinate from glucose as one of the substrates for benzylsuccinate synthesis, we opted to implement the *bbs* operon of *G. metallireducens* as the source of the β-oxidation genes including the appropriate ETF and EMO (Figure 4).

The generation of the second substrate for benzylsuccinate synthesis, benzoyl-CoA is of concern, because it is no naturally occurring metabolite of *E. coli*. To enable the cells to obtain benzoyl-CoA from benzoate added to the medium, we established two separate synthetic modules for the uptake and activation of benzoate to benzoyl-CoA by combining the genes for the benzoate transporter BenK with those of either a benzoate-CoA ligase or a CoA-transferase. These enzymes are expected to activate benzoate to the CoA-thioester at different energy expense: benzoate-CoA ligase generates AMP, whose regeneration consumes two ATP equivalents, while the CoA-transferase utilizes succinyl-CoA, which is regenerated with consumption of only one ATP equivalent. Notably, the genes for transporter and activating enzymes were only tolerated on a common plasmid in *E. coli* when we included additional genes for benzoyl-CoA-consuming enzymes. We tried this either with the *bbsABCD* genes, which represent the last two enzymes of the β-oxidation pathway of benzylsuccinate (Figure 1) or the *bis* gene coding for a biphenyl synthase in the rowan tree *Sorbus acuparia*, which synthesizes the secondary plant product 3,5-dihydroxybiphenyl from benzoyl-CoA and three malonyl-CoA [46,47]. The observation that the presence of additional genes coding for either type of benzoyl-CoA utilizing enzyme allowed the construction of the plasmids corroborates our hypothesis that the introduced metabolic module of a benzoate transporter and activating enzyme into *E. coli* depleted the cellular CoA pool. Although no benzoate had been added during these experiments, benzoate derivatives are known as minor components of yeast extract [48] and therefore available in the rich media used. Because we did not observe any negative effects from the presence of the *bbsABCD* genes in the plasmids and planned to combine it with the full *bbs* operon, we continued to work with these plasmids for the following steps. The presence of double copies of these genes in the production strains also did not cause any apparent problems.

After combining either type of benzoyl-CoA synthetic module with the β-oxidation enzymes from *G. metallireducens*, we immediately observed the production of benzylsuccinate in the supernatants of aerobic cultures supplied with succinate and benzoate. However, the concentrations of 0.3 to 1.7 nM obtained in these experiments were very low and close to the detection limit, and the effects of CoA ligase vs. CoA-transferase for benzoate activation appeared to differ between minimal and rich media. After shifting to anaerobic conditions, benzylsuccinate production yields increased strongly, although only benzoate was fed to the medium, and succinate needed to be produced as a fermentation product from glucose (Figure 5B). Since the cultures containing the CoA ligase module performed much better in benzylsuccinate production, only they were used for further optimization studies. These included the introduction of a mutant *mscS* gene for an unspecific export protein, which increased benzylsuccinate concentration in the supernatant by more than 3-fold under fermentation conditions. Finally, growth under fumarate respiration conditions resulted in a 10-fold increase in benzylsuccinate yield, compared to fermentation conditions. The observed decrease of product yield with fumarate respiration in the presence of MscS may be explained by the adverse effects of the exporter on maintaining the proton gradient necessary for ATP regeneration via fumarate respiration. The observed drop of intracellular benzylsuccinate concentration in fumarate-respiring, but not in fermenting cells when MscS was introduced, may either indicate different exporter efficiencies at the different growth conditions or a lower production rate in the presence of MscS. In the absence of any refinement procedures such as optimizing the medium composition, incubation time, temperature, pH, or other parameters, we regard the maximum yield of 4.8 µM benzylsuccinate (equal to ca. 1 mg/L) observed in this study as a promising first step to develop a viable process, which might be possible after a further 1000-fold yield improvement.

Further possible steps to improve the yield of benzylsuccinate produced are clearly necessary. Our initial studies indicate that one of the major bottlenecks is the lack of a proper export system for benzylsuccinate since the measured intracellular concentrations of benzylsuccinate were more than 10-fold higher than those excreted in the medium. One of the reasons involved in higher productivity under anaerobic conditions may be the induction of C4-dicarbonic acid transporters in *E. coli* [49] which may also take part in benzylsuccinate export. Potentially more dedicated exporter candidates to be evaluated in further optimization attempts may be the products of conserved “solvent stress” operons encoded in the genomes of many anaerobic toluene degraders, one of which is actually co-induced with the *bss* and *bbs* operons in toluene-degrading *A. aromaticum* [50]. Further improvements may result from using succinate-overproducing strains of *E. coli* [51] which would allow access to much higher substrate concentrations and less unwanted fermentation products formed from glucose, or switching to other naturally succinate-overproducing species such as *Basfia succiniproducens* [52].

The observed production of benzylsuccinate under fermentative and fumarate-respiratory conditions may be rationalized by some thermodynamic considerations. Succinate is one of the usual, albeit minor fermentation products of *E. coli* [53], and its production consumes both NADH equivalents generated during glycolysis per fermented glucose. Because the synthesis of succinate branches off from the high-energy intermediate phospho-enol-pyruvate (PEP), only one ATP is conserved from the respective glucose molecule during glycolysis (Figure 7A). The acetyl-CoA generated from pyruvate by pyruvate-formate lyase (PFL) is converted to acetyl-phosphate which allows for the conservation of a second ATP (Figure 7A). Concomitantly, PEP is converted to oxaloacetate with the CO_2_ generated from formate cleavage via formate:hydrogen lyase, which is converted to succinate via the reductive branch of the citric acid cycle (Figure 7A). This includes the reaction of fumarate reductase, which establishes a redox loop with NADH:quinone oxidoreductase and conserves the energy difference between the NADH/NAD^+^ and the succinate/fumarate redox couples in the form of a proton gradient. The synthetic benzylsuccinate-generating pathway should allow the addition of further reductive steps to this cascade, starting with succinate, which is activated by the CoA-transferase BbsEF [23], covalently bound to a benzoyl-CoA cosubstrate and then enters reductive reverse β-oxidation (Figure 7). To obtain an overview of the thermodynamic feasibility of the proposed pathway, calculations were performed using ΔG_f_°’ values of the molecules involved from [54], except for benzylsuccinate, whose ΔG_f_°’ value was calculated as −547 kJ mol^−1^ from the results of a QM simulation of the benzylsuccinate synthase reaction [29]. Using Equation (1), we calculated a free enthalpy change of −262 kJ mol^−1^ for the formation of acetate, H_2_, and succinate from glucose, which occurs during mixed acid fermentation. The value correlates well to the three ATP equivalents (one ATP equal to values of 72–80 kJ mol^−1^) predicted to be formed during the pathway via glycolysis, acetate kinase, and fumarate reductase (Figure 7A).
glucose → acetate^−^ + H_2_ + succinate^2−^ + 3 H^+^ ΔG°’ = −262 kJ mol^−1^(1)

If benzylsuccinate formation is considered in a balanced equation with glucose degradation to acetate; H_2_ and CO_2_, the available energy value increases according to Equation (2), but only marginally relative to one glucose molecule. However, the formation of benzylsuccinate as an additional fermentation product may allow additional recycling of the released hydrogen or formate molecules from glucose fermentation by coupling either hydrogenase-2 [55,56] or formate dehydrogenase-N or -O of *E. coli* [57,58] with fumarate reductase and EMO. Under these conditions, the energy yield value per glucose can become significantly higher and may approach up to four ATP equivalents per glucose as shown in Equation (3). Therefore, benzylsuccinate production may actually be beneficial for the anaerobic growth of *E. coli*, and the additional costs for benzoate transport and activation should be covered by the additional gain of energy.
2 glucose + benzoate^−^ + H^+^ → 3 acetate^−^ + 3 H_2_ + 2 CO_2_ + benzylsuccinate^2−^ + 5 
H^+^ ΔG°’ = −546.6 kJ mol^−1^ or −273.3 kJ (mol glucose)^−1^(2)
5 glucose + 4 benzoate^−^ + 4 H^+^ → 6 acetate^−^ + 2 CO_2_ + 4 benzylsuccinate^2−^ + 6 
H_2_O + 14 H^+^ ΔG°’ = −1577.3 kJ mol^−1^ or −315.5 kJ (mol glucose)^−1^(3)

Concluding this study, we show that it is possible to use degradative β-oxidation pathways in reverse for biosynthetic purposes. In addition to the *bbs* genes used in this study, almost identical operons are known to be involved in the degradation of succinate adducts of other hydrocarbons, such as 2-methylnaphthalene, cresols, or xylenes [4,59]. In addition, many other natural products are degraded by β-oxidation. Therefore, the reversal of known degradation pathways may be an interesting concept to produce such molecules in an analogous manner as shown here. In recent years, several unrelated β-oxidation pathways have already been used to construct synthetic pathways for easy access to secondary plant compounds or other interesting metabolites [60]. As a second building block, we established a simple metabolic module for the synthesis of benzoyl-CoA, which does not naturally occur in *E. coli*. Similar uptake and activation modules are conceivable for many other substrates, enabling biosynthetic routes for more complicated derivatives without the need to establish a more complex pathway for synthesizing the activated molecule from scratch. Since benzoyl-CoA is used as an intermediate in many different biosynthetic pathways, just the two modules described here may be combined with other enzymes to produce a variety of additional target compounds.

## 4. Materials and Methods

### 4.1. Cultivation of Bacteria

*Aromatoleum aromaticum* (DSMZ 19018) and *Geobacter metallireducens* (DSMZ 7210) were grown under denitrifying conditions in minimal media with benzoate as substrate as described previously [25,27]. *Eschericha coli* strains were routinely grown at 37 °C in LB medium or M9 minimal medium [61], which were supplemented with substates or antibiotics as needed. In order to induce the added genes, AHT (0.2 mg mL^−1^) and IPTG (0.5 mM) were added at OD_578_ values of 0.4 to 0.6, and the cells were subsequently incubated at 15 °C. The media used for these experiments were autoinduction medium [62] (ZYP-5052, VWR life science, Radnor, PA, USA), TGYEP [63], LB or M9 minimal medium with 10 mM glucose. For producing recombinant enzymes, the induction phase lasted for 16 h, while the benzylsuccinate-producing cultures were incubated for 3 days. Growth was monitored by following the increase in OD_578_.

### 4.2. Molecular Biological Techniques

Chromosomal DNA was prepared as described in [64], and plasmids were prepared using the GeneJet kit (Fermentas, St. Leon-Rot, Germany). PCR reactions were performed using the DNA primers listed in Table A1. DNA fragments were electrophoretically separated in 1% agarose gels in TAE buffer and stained with ethidium bromide. Restriction and ligation assays were performed under standard conditions [65] as recommended by the suppliers. Plasmids or ligation mixtures were transformed into competent *E. coli* cells, which were prepared according to Inoue et al. (1990) [66]. DNA sequencing was performed by SeqLab (Göttingen, Germany).

### 4.3. Combinatory Cloning

The expression vectors containing multiple genes in the form of an artificial operon were produced by combinatory cloning according to the instructions of the Stargate cloning system (IBA, Göttingen, Germany). To this end, the genes of interest were initially amplified by Phusion DNA polymerase (Finnzymes, Espoo, Finland) and cloned into the pEntry donor vector (pE-IBA20), using appropriately engineered recognition sites of the type IIS restriction enzyme LguI in the PCR primers used (Table A1). After insertion of the DNA fragment, the flanking LguI recognition sites are lost and replaced by Esp3I sites provided by the donor vector. In a second step, the genes are then transferred to the actual expression vectors (e.g., the pASG series) by oriented insertion of the Esp3I fragments. The expression vectors provide an anhydrotetracycline-inducible tet promotor and optional N- or C-terminal strep II-affinity tags. For constructing synthetic operons, the Esp3I fragments containing the genes to be combined were transferred from the original donor vectors into vectors pNFUSE and pCFUSE, respectively. These vectors provide an intergenic region between the genes to be combined, and the genes are again flanked by LguI restriction sites. Cutting the insert out by LguI allows the construction of a new donor vector in pEntry containing the fused genes between Esp3I restriction sites, which can then be transferred to the appropriate expression vector. The same procedure has been repeated to create the multiple fusion constructs in the expression plasmids pLigBen, pTransBen, and pLigBenEx (Figure 4).

### 4.4. Cloning of the bbs Operon

Using a Long-Range PCR method with KOD Hot Start DNA polymerase (biotechrabbit, Berlin, Germany), the genes of the entire *bbs* operon of *G. metallireducens* as well as the broad-host-range vector pBBR2 [41] were amplified, and ligated together, producing an expression plasmid of 16.3 kb containing the *bbs* operon behind an IPTG-inducible *lac* promotor (pBeta, see Figure 4).

### 4.5. Enzyme Purification

To obtain recombinant enzymes, cultures of 2 l of *E. coli* Rosetta (DE3) pLysS containing either the N-terminally strep-tagged *bclA* or *bct* gene on an AHT inducible plasmid were grown to an OD_578_ of 0.5, induced by adding AHT and incubated for further 16 h at 15 °C. The cells were harvested by centrifugation, suspended in 30 mL buffer A (10 mM Tris/HCl pH 8, 2 mM MgCl_2_, 2 mM dithiothreitol), and lysed by passage through a French Press cell. In the case of benzoate-CoA ligase, the cell extract was then subjected to a fractionated ammonium sulfate precipitation to obtain the fraction precipitating between 33%, and 60% saturation. The precipitate was retrieved by centrifugation, resuspended in buffer A, and desalted by a passage through a PD10 column (6 mL; GE Healthcare, Freiburg, Germany) using buffer A. Finally, the protein was applied to a UnoQ sepharose column (BioRad, München, Germany) and eluted by a gradient of 50–500 mM KCl in buffer A over 20 column volumes. The fractions were collected and tested for benzoate-CoA ligase activity. The succinyl-CoA:benzoate CoA-transferase was purified from the respective extract in one step, using affinity chromatography on a 5 mL streptactin column as indicated by the supplier (IBA, Göttingen, Germany).

### 4.6. Enzyme Activity Assays

The activity of benzoate CoA ligase was determined by a coupled photometric assay as described previously, coupling the production of AMP from ATP to the oxidation of 2 NADH via added myokinase, pyruvate kinase, phospho-*enol*-pyruvate, and lactate dehydrogenase [34,67]. Succinyl-CoA:benzoate CoA-transferase activity was measured using the HPLC-based detection of the respective thioesters. To this end, 30 µg enzyme was incubated for 10 min at 30 °C in 200 µL of buffer B (50 mM MES pH 6.2, 5 mM MgCl_2_) with 1 mM benzoate and 250 µM succinyl-CoA. The reaction was stopped by adding 20 µL of a 2 M NaHSO_4_ solution and incubation on ice. After removing the precipitated proteins by centrifugation, the supernatants were analyzed by HPLC, using an RP-18 column (Varian Microsorb 100–5; 150 × 4.6 mm) which was developed over 20 min at a flow rate of 1 mL min^−1^ and a gradient of 3–20% acetonitrile in 50 mM MES buffer (pH 6.2). CoA thioesters were detected by their absorption at 260 nm via a diode array detector. The retention times of succinyl-CoA and benzoyl-CoA standards were at 2 and 10 min, respectively.

### 4.7. Extraction and Detection of Benzylsuccinate

Detection and quantitation of produced benzylsuccinate were achieved by HPLC-MS analytics. To this end, the induced cultures were separated into cell mass and supernatants by centrifugation. Subsequently, either the supernatants or both fractions (as well as the benzylsuccinate standard solutions) were spiked with known amounts of phenylsuccinate as internal standard. The cell pellets were washed by suspending and re-centrifuging them in 1 mL 100 mM Tris-HCl buffer (pH 7.5), then they were resuspended in 900 µL water containing 100 µM phenylsuccinate, acidified to pH 2 by adding 100 µL trifluoroacetic acid (TFA) and heated to 60 °C for 10 min. The supernatants and the standard solutions were likewise spiked with phenylsuccinate (at 2 µM, which should be concentrated to 100 µM after extraction) and acidified with TFA. The organic acids from the acidified supernatants and cell pellet fractions were then extracted twice with equal volumes of ethyl acetate (50 mL for supernatants, 1 mL for cell pellets). After combining the organic phases, the solvent was evaporated and the residues were solved in 20% acetonitrile containing 0.1% TFA. These samples as well as the benzylsuccinate standards spiked with phenylsuccinate were then separated by isocratic HPLC-MS runs over an RP-18 column (3 µm, 150 × 4.6 mm), using 20% acetonitrile/0.1% TFA as solvent at a flow rate of 0.5 mL min^−1^ on an 1100 HPLC system (Agilent). Mass spectrometric detection was achieved by coupling an LTQ-FT Ultra FT-ICR mass spectrometer (ThermoFisher Scientific; Darmstadt, Germany) to the outlet of the HPLC column. Masses were measured in negative ion mode. Extracted ion chromatograms of benzylsuccinate (C_11_H_12_O_4_; [M-H]^−^ = 207.0665 *m*/*z*) and phenylsuccinate (C_10_H_10_O_4_; [M-H]^−^ = 193.0507 *m*/*z*) were generated and the corresponding signals were quantified by integration. Benzylsuccinate showed a retention time of 17.2 min, and phenylsuccinate of 19.8 min (Figure A3). The internal standard phenylsuccinate was used for the normalization of the different extractions and HPLC-MS runs. Normalized benzylsuccinate concentrations were converted to the supernatant and cytoplasmic values based on the concentrating factor from the extraction procedures and a volume (dry cell mass)^−1^ ratio of 1 mL (0.4 g dry mass)^−1^ [68].

### 4.8. Other Techniques

Protein concentrations were determined by the Coomassie-binding assay using bovine serum albumin as standard [69], and proteins were visualized by SDS-polyacrylamide gel electrophoresis using gels with 10–12% polyacrylamide concentrations, which were stained with coomassie brilliant blue R-250 [70].

## Figures and Tables

**Figure 1 molecules-29-00415-f001:**
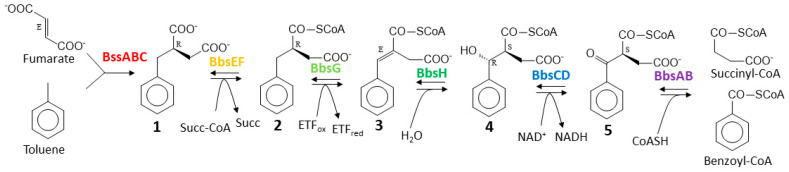
Pathway of anaerobic toluene degradation. Enzymes involved are benzylsuccinate synthase (BssABC), benzylsuccinate CoA-transferase (BbsEF), benzylsuccinyl-CoA dehydrogenase (BbsG), benzylidenesuccinyl-CoA hydratase (BbsH), (α-hydroxybenzyl)succinyl-CoA dehydrogenase (BbsCD), and benzoylsuccinyl-CoA thiolase (BbsAB). The intermediates and their known or inferred stereochemical conformations are (*R*)-benzylsuccinate (**1**), (*R*)-2-benzylsuccinyl-CoA (**2**), (*E*)-2-benzylidenesuccinyl-CoA (**3**), (*S*,*R*)-2-(α-hydroxybenzyl)succinyl-CoA (**4**), and (*S*)-2-benzoylsuccinyl-CoA (**5**).

**Figure 2 molecules-29-00415-f002:**
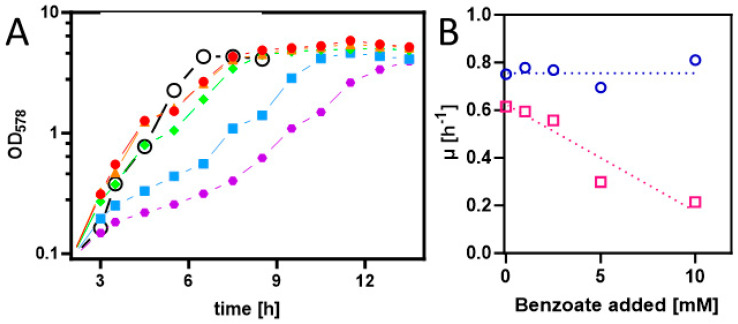
(**A**) Growth curves of *E. coli* Rosetta(DE3) pLysS containing expression plasmid pASG_mod-*benK* in LB medium with different added benzoate concentrations (0, 1, 2.5, 5, and 10 mM in red, orange, green, blue, and purple). Gene induction was started 3 h after inoculation. The growth curve of the same strain without a plasmid in LB medium with 5 mM benzoate is shown (open black circles). (**B**) Growth rate dependence of *E. coli* containing or lacking the *benK* gene from added benzoate (magenta and blue, respectively).

**Figure 3 molecules-29-00415-f003:**
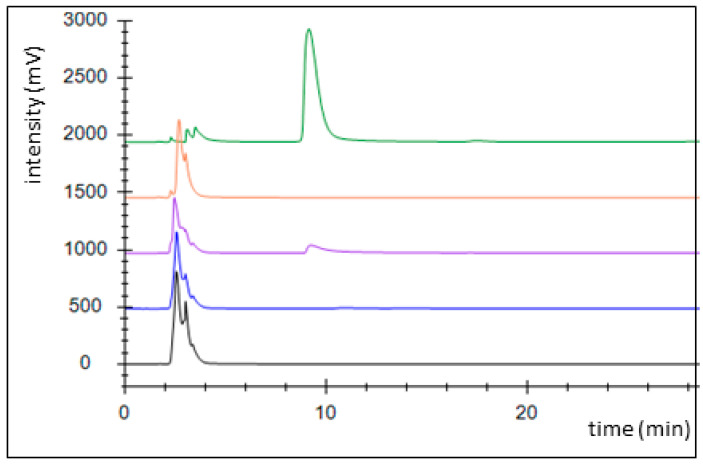
Activity assessment of succinyl-CoA:benzoate CoA-transferase by HPLC analysis. Chromatograms: control without added enzyme (black), full assay at 0 (blue), and 10 min of incubation (purple), succinyl-CoA standard (1 mM; orange), benzoyl-CoA standard (1 mM, green).

**Figure 4 molecules-29-00415-f004:**
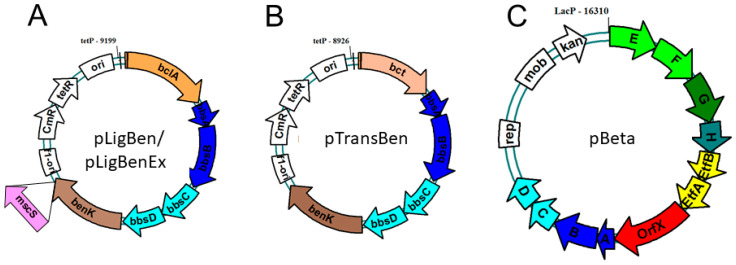
Expression plasmids constructed for benzylsuccinate production. Both pLigBen (**A**) and pTransBen (**B**) contain a ColE1 replication origin, while pBeta (**C**) contains a broad range origin of the pBBR plasmids. Letters refer to the contained *bbs* genes. Plasmid pLigBenEx was constructed by adding a mutant *mscS* gene as indicated.

**Figure 5 molecules-29-00415-f005:**
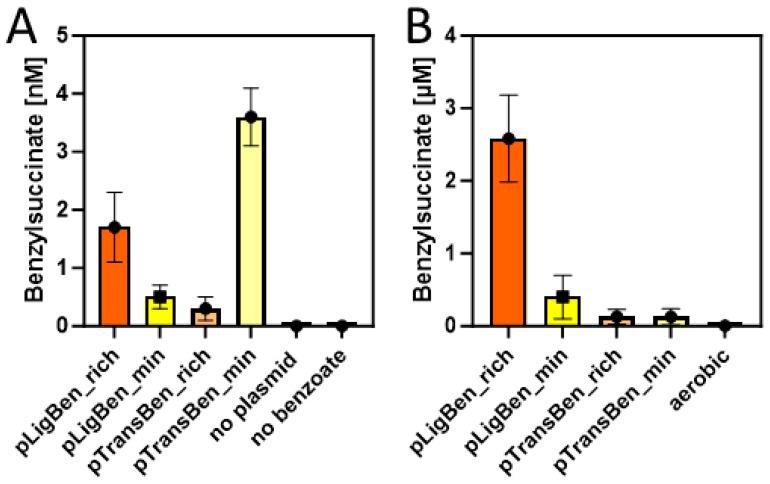
Benzylsuccinate yields in the culture supernatants of cells carrying plasmid pBeta and either pLigBen or pTransBen after three days of incubation at 15 °C in rich (orange) and minimal media (yellow). (**A**) Experiments under aerobic conditions with added succinate and benzoate. Results of control experiments without the plasmids or without added benzoate are included. (**B**) Experiments under anaerobic conditions with added glucose and benzoate. The experiment from (**A**) yielding the most product under aerobic conditions is added to illustrate the different scales of the graphs.

**Figure 6 molecules-29-00415-f006:**
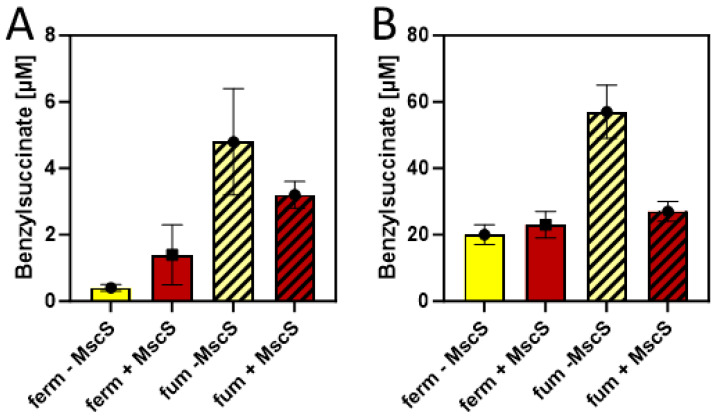
Benzylsuccinate yields of cells carrying plasmids pBeta and pLigBen after anaerobic incubation for three days at 15 °C in minimal media. The addition of an MscS exporter is indicated by shifting from yellow to brown columns, and fumarate-respiring cultures are indicated by hatching. (**A**) Product concentrations in culture supernatants. (**B**) Intracellular product concentrations.

**Figure 7 molecules-29-00415-f007:**
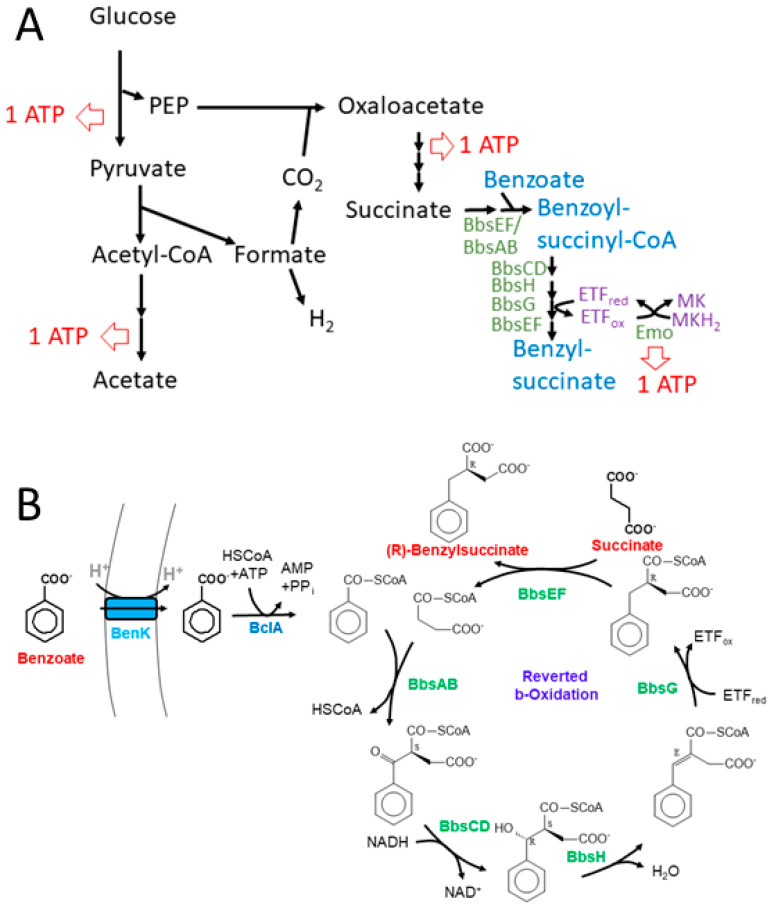
Representation of the expanded benzylsuccinate-producing fermentation pathway. (**A**) Scheme of the acetate- and succinate-forming branches of mixed acid fermentation including the added module for benzylsuccinate formation. Possible ATP regeneration sites are indicated. (**B**) Detailed scheme of the synthetic pathway of benzoyl-CoA-producing and β-oxidation modules involved in benzylsuccinate synthesis.

**Table 1 molecules-29-00415-t001:** Purification of recombinant benzoate-CoA ligase.

Purification Step	Volume (mL)	Protein (mg)	Activity (U)	Specific Activity (U mg^−1^)	Yield	Enrichment
Cell extract	12	365	2412	6.6	100%	1
(NH_4_)_2_SO_4_-precipitation (33–60% saturation)	2.5	80	1593	20	66%	3
PD10 fraction	2.0	58	1124	19	47%	3
UnoQ fraction	3.0	2.6	274	107	11%	16

## Data Availability

Data are contained within the article.

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
