# Peer review of "A Synthetic Pathway for the Production of Benzylsuccinate in Escherichia coli"

_molecules, 2024, doi:10.3390/molecules29020415_

Round 1

Reviewer 1 Report

Comments and Suggestions for Authors

The paper was interesting and I have only a few comments, listed below. In particular, the authors might pay attention to the comments about insufficient comparison to the alternative chemical synthesis, the cost of benzylsuccinate, and by how much the productivity of the E. coli based pathway would need to be improved to make it industrially valuable. While it may not be possible to provide all of this information, I think attempting to do so would frame the work to demonstrate the potential of synthetic biology to produce compounds that are hard to produce in enantiomerically pure form by chemical synthesis. The quality/resolution of the figures must be improved. 

List of comments

The abstract does not explain the interest in benzylsuccinate. Please state the isomer that is formed by the pathway in the abstract. 

Line 35

“reported by relatively complex procedures [9] which yield racemic mixtures”: some detail on the complexity, number of steps, cost, etc. would help appreciate the need for a biological synthetic pathway. From a quick search it appears that the compound is not commercially available and would probably need to be synthesized on demand. Please discuss whether this lack of availability is because the chemical synthesis is not worth it or whether the demand for it is lower than the introduction of the paper may lead one to believe.

Line 121

“Because purification attemots” 

Line 130

Table 1: Text for column headers too large, text does not fit

Line 157

“activities may result in excessive formation of benzoyl-CoA and depletion of the 157 cytoplasmic CoA pool”: It is unclear whether benzoate was added under these conditions, although it sounds like it was. Could the authors comment on how stable this construct was in the absence of benzoate? 

Line 178

Italicize G. metallireducens

Line 271

The quality of figures (resolution) is poor

Line 279

Compound number 5 is missing 

Line 296

Figure 4; It would be appreciated if the authors could add the sequences for the final plasmids to the supporting information

Line 301

pTtansBen

Line 307

Figure 6 

Lines 367 to 371 

As for an earlier comment, it is not clear whether benzoate was included in media used for these experiments. The mode of toxicity proposed is very reasonable (see e.g., https://doi.org/10.3109/03602532.2014.908903). However, if this phenotype is observed in the absence of benzoate added to the media, it raises interesting questions (are there traces of benzoate in the media, are other organic acids activated, leading to the same problem, etc.). 

Lines 386 to 388 

Here it may be valuable to discuss why significant reduction in intracellular concentrations was not observed. 

Lines 395 to 396

It would add value to the paper if the authors could discuss the commercial value and availability of benzylsuccinate, and how much the productivity of their pathway would need to be increased for it to be commercially competitive. 

Line 406

Please check “the bss and bbs operons” 

Line 415

Please check “deviates branches off” 

Line 504

“anhydrotetracycline”

Line 512

“xpression” 

Line 533

“Succinyl-CoA:benzoate CoA-transferase. The CoA-transferase was purified from”

Line 543

“1 mM benzoate and 250 mM”; please check that this should not be µM 

Lines 605 and 609

“indiceted”

Author Response

List of comments

The abstract does not explain the interest in benzylsuccinate. Please state the isomer that is formed by the pathway in the abstract. 

Isomer has been added in Abstract and Introduction

Line 35

“reported by relatively complex procedures [9] which yield racemic mixtures”: some detail on the complexity, number of steps, cost, etc. would help appreciate the need for a biological synthetic pathway. From a quick search it appears that the compound is not commercially available and would probably need to be synthesized on demand. Please discuss whether this lack of availability is because the chemical synthesis is not worth it or whether the demand for it is lower than the introduction of the paper may lead one to believe.

We have added a few more references for newer synthetic procedures and stated the need of at least four steps. Furthermore, we added a sentence on the cost of benzylsuccinate in comparison to benzoate and succinate, which starts at about 1000-fold higher for racemic BS (looked up and compared from the same companies and in comparable purities, since the prices of benzoate batches alone vary by more than 100-fold depending on purity and bulk). I think that should satisfy the readers, since the detailed synthesis procedures are available in the references and the prices can easily be checked from the online chemical catalogues.

Line 121

“Because purification attemots”  has been corrected

Line 130

Table 1: Text for column headers too large, text does not fit

Table format has been warped in the reviewers copies. It will be submitted as separate file in revision

Line 157

“activities may result in excessive formation of benzoyl-CoA and depletion of the 157 cytoplasmic CoA pool”: It is unclear whether benzoate was added under these conditions, although it sounds like it was. Could the authors comment on how stable this construct was in the absence of benzoate? 

We have not added benzoate to the media while constructing the plasmids, but expect it to be present to some extent in the rich media used (see added ref). Anyway, we only obtained these plasmids when we had added either the bbsABCD genes of bis, which we interpreted as requirement to have a benzoyl-CoA-converting activity present.

Line 178

Italicize G. metallireducens has been corrected

Line 271

The quality of figures (resolution) is poor

The figures (along with the rest of the text) have been converted to pdf format for the review process, but they will be re-submitted as high-resolution files with the revised paper

Line 279

Compound number 5 is missing 

No. 5 has been added

Line 296

Figure 4; It would be appreciated if the authors could add the sequences for the final plasmids to the supporting information

For the purpose of re-constructing the plasmids, we included the sequences of all PCR primers used to amplify the genes and create the appropriate intergenic sequences. Since all genes can easily be located in the DNA data base by their indicated designations, we do not think that a posting of the final constructs is necessary, which would consume large storage space even as supplement.

Line 301

pTtansBen has been corrected

Line 307

Figure 6 has been corrected

Lines 367 to 371 

As for an earlier comment, it is not clear whether benzoate was included in media used for these experiments. The mode of toxicity proposed is very reasonable (see e.g., https://doi.org/10.3109/03602532.2014.908903). However, if this phenotype is observed in the absence of benzoate added to the media, it raises interesting questions (are there traces of benzoate in the media, are other organic acids activated, leading to the same problem, etc.). 

we actually have observed this without added benzoate and assumed that enough benzoate or analogs are present in the rich media to cause this effect (see added reference on the presence of 2,3-dihydroxybenzoate in yeast extract). After we had our plasmid constructs, we did not re-analyse the effect because it is unrelated to the topic of this study.

Lines 386 to 388 

Here it may be valuable to discuss why significant reduction in intracellular concentrations was not observed. 

We added a sentence dealing with potential explanations of the different effects of the exporter on intracellular concentrations under fermentative and fumarate-respiring conditions. However, nothing definitive is known at this point, so we do not want to add too much speculations.

Lines 395 to 396

It would add value to the paper if the authors could discuss the commercial value and availability of benzylsuccinate, and how much the productivity of their pathway would need to be increased for it to be commercially competitive. 

We added a statement what yield improvement we consider to make the process viable

Line 406

Please check “the bss and bbs operons” 

names are OK

Line 415

Please check “deviates branches off” has been correcetd

Line 504

“anhydrotetracycline” has been corrected

Line 512

“xpression” has been corrected

Line 533

“Succinyl-CoA:benzoate CoA-transferase. The CoA-transferase was purified from”

has been corrected. Thanks for spotting the error.

Line 543

“1 mM benzoate and 250 mM”; please check that this should not be µM 

the value for succinyl-CoA is indeed wrong. It has been corrected to 250 µM. 1 mM benzoate is correct.

Lines 605 and 609

“indiceted” has been corrected

Reviewer 2 Report

Comments and Suggestions for Authors

Experiments were well-designed and a manuscript is well-written. Also novelty of their work is high. This work can be acceptable as is. However, conclusion section is lacking. Please provide proper conclusion based on research results.

In this work, toluene degradation pathway to generate benzylsuccinate was investigated. This pathway showed 100-fold product yield compared to aerobic process. My detail comments and suggestions are follows.

- Line 125, use full word as Table.

- In a text, Fig, 6 is mentioned, however Fig. 6 is not provided. Check miss-writting.

- A section of conclusion is empty. I recommend to provide conclusion in the text, since your discussion is so long. 

Author Response

In this work, toluene degradation pathway to generate benzylsuccinate was investigated. This pathway showed 100-fold product yield compared to aerobic process. My detail comments and suggestions are follows.

- Line 125, use full word as Table. has been changed

In a text, Fig, 6 is mentioned, however Fig. 6 is not provided. Check miss-writting.

Fig. 6 was wrongly labeled as a (second) Fig. 5. Thanks for notifying us.

 A section of conclusion is empty. I recommend to provide conclusion in the text, since your discussion is so long.

The “conclusion” heading was left over from the blank form provided, and I just forgot to delete it. Since a conclusion is not mandatory, I had devised the discussion in a way to include the conclusions. Therefore, I deleted the subheading in the revision.

Reviewer 3 Report

Comments and Suggestions for Authors

In their paper, Mock et al. describes the design of the synthetic pathway for the production of benzylsuccinate from succinate and benzoate in Escherichia coli using reverted beta-oxidation.

I recommend publication of this paper in Molecules after minor corrections have been made.

1) Please add the molecular formulas of benzylsuccinate and phenylsuccinate to the "Extraction and detection of benzylsuccinate" section, line 570 (C11H12O4 and C10H10O4 with subscripts).

2) Benzylsuccinate and phenylsuccinate has two carboxyl groups and can form double charged ions [M-2H]2- with m/z 103.029 and 96.525 in mass spectra. I don't observe 96.525 in the phenylsuccinate spectrum (bottom panels of figure A3), which may be caused by its low m/z value. If you observe 103.029 in the benzylsuccinate spectrum, please, add this ion to the "Extraction and detection of benzylsuccinate" section.

3) "L" instead of "l" should be used to abbreviate liter. "mL" instead of "ml" should be in Table 1. Same for lines 139, 395, 478, 521, 524, 529, 534, 542, 543, 546, 556, 557 twice, 562 twice, 567, 577.

4) Table 1. -1 should be superscript for "U mg-1".
5) Line 577. -1 should be superscript for "(0.4 g dry mass)-1".

Author Response

1) Please add the molecular formulas of benzylsuccinate and phenylsuccinate to the "Extraction and detection of benzylsuccinate" section, line 570 (C11H12O4 and C10H10O4 with subscripts).

Has been added

2) Benzylsuccinate and phenylsuccinate has two carboxyl groups and can form double charged ions [M-2H]2- with m/z 103.029 and 96.525 in mass spectra. I don't observe 96.525 in the phenylsuccinate spectrum (bottom panels of figure A3), which may be caused by its low m/z value. If you observe 103.029 in the benzylsuccinate spectrum, please, add this ion to the "Extraction and detection of benzylsuccinate" section.

We have shown a zoom view of the m/z signals of the mono-anions to shown that they were clearly detectable in the assays. The positions of the [M-2H]2- ions are not contained, but we made sure that authentic compounds were eluting at the peak positions by comparing their full MS spectra with those of the reference compounds. Because only the mono-anion peaks were used for quantitation of the compounds, we believe that a representation of the full MS spectra is not necessary.

3) "L" instead of "l" should be used to abbreviate liter. "mL" instead of "ml" should be in Table 1. Same for lines 139, 395, 478, 521, 524, 529, 534, 542, 543, 546, 556, 557 twice, 562 twice, 567, 577.

The “ml” and "µl" values have been changed to “mL” and "µL" throughout

4) Table 1. -1 should be superscript for "U mg-1". has been corrected
5) Line 577. -1 should be superscript for "(0.4 g dry mass)-1". has been corrected